# Learning and Reasoning with Model-Grounded Symbolic Artificial Intelligence Systems

**Aniruddha Chattopadhyay**                    STUDYANIRUDDHA@GMAIL.COM
**Raj Dandekar**                                          RAJD@MIT.EDU
**Kaushik Roy**                                           KROY2@UA.EDU

**Editors:** Leilani H. Gilpin, Eleonora Giunchiglia, Pascal Hitzler, and Emile van Krieken

## Abstract

Neurosymbolic artificial intelligence (AI) systems combine neural network and classical symbolic AI mechanisms to exploit the complementary strengths of large-scale, generalizable learning and robust, verifiable reasoning. Numerous classifications of neurosymbolic AI illustrate how these two components can be integrated in distinctly different ways. In this work, we propose reinterpreting instruction-tuned large language models as **model-grounded symbolic AI systems**—where natural language serves as the symbolic layer, and grounding is achieved through the model's internal representation space. Within this framework, we investigate and develop novel learning and reasoning approaches that preserve structural similarities to traditional learning and reasoning paradigms. Comprehensive evaluations across complex mathematical reasoning procedures of varying difficulties provide insights into the effectiveness of our approach towards learning efficiency and reasoning reliability.

## 1. Introduction

Neurosymbolic AI has sought to combine the powerful and generalizable learning capabilities of neural networks with the explicit and verifiable reasoning abilities in symbolic systems. This "hybrid" approach has gained renewed attention in recent times as a way to overcome limitations of large language models in complex reasoning tasks Fang et al. (2024); Sheth et al. (2024). Large language models demonstrably struggle with logical consistency, abstraction and adapting to new concepts or scenarios beyond their training distribution Capitanelli and Mastrogiovanni (2024). Integrating symbolic knowledge and reasoning is seen as a promising avenue to enhance large language model capabilities towards enabling AI systems to leverage both data-driven learning and high-level knowledge representation Colelough and Regli (2024). Various studies have demonstrated the core challenge with neurosymbolic AI systems lies in working out the mathematical framework for achieving **unified symbol grounding** – bridging symbols grounded in discrete explicit knowledge representations with symbols grounded in implicit continuous abstract vector spaces Xu et al. (2022); Wagner and d'Avila Garcez (2024). We refer to the latter form of grounding as **model grounding**. Traditionally, the symbol grounding problem involves linking symbols to their real-world referents (usually explicitly through a knowledge representation such as a knowledge graph) Harnad (1990). Empirical evidence suggests that large language models may lack sufficient capabilities for such grounding, particularly in real-world contexts Bisk et al. (2022). However, recent studies offer alternative perspectives. Recent work has argued that the symbol grounding problem may not apply to large language models as

previously thought by stating that grounding in *pragmatic norms* (grounding in abstract vector spaces) is sufficent for obtaining robust task solutions achieved via language model reasoning Gubelmann (2024). Other works have proposed that instruction-tuned large language models (e.g., by reinforcement learning from human feedback) confer intrinstic meaning to symbols through grounding in vector spaces Chan et al. (2023).

In this work, we re-interpret instruction-tuned large language models as symbolic systems in which the symbols are natural language instructions that have *model-grounding* in the model's internal representations and propose a novel learning regime:

---

**Model-grounded Symbolic Learning Perspective**

*Can we conceptualize task-learning by large language models (LLMs) as an iterative learning process through a training dataset where symbolic natural language-based interactions characterize each training run influencing model behavior?*

**Rather than viewing learning solely as parameter-state updates via gradient descent, we interpret it as refining the LLM's *task functionality state*—a prompt plus a structured memory of critiques. These iterative refinements arise from repeated training dataset interactions and an external judge that identifies prompt errors (or gradients), facilitating "learning" towards task objectives.**

---

Under such a learning regime, there are direct analogs to the traditional learning setup – train-validation-test splits, number of training runs through the dataset (epochs), gradient accumulation (when to trigger a prompt revision), model saving (saving the history of interactions and prompt-revisions, along with the final prompt), and model loading at inference time on the test set (loading the same history and warm starting with a few sample generations before testing begins). We test our approach on a suite of mathematical reasoning procedures of varying complexity. Comprehensive experiments provide insights into the effectiveness of our proposed method and framework. Thus our main contributions are:

---

**Main Contributions:**

- **Model-Grounded Symbolic Framework.** Treat instruction-tuned LLMs as symbolic systems, with natural language as symbols grounded in the model's internal representations.

- **Iterative Prompt-Refinement.** Introduce a structured approach to "learning" via iterative prompt revisions and critique-driven updates, bridging symbolic reasoning and gradient-based optimization.

- **Empirical Validation.** Demonstrate improved reasoning reliability, adaptability, and sample efficiency across mathematical reasoning tasks.

---

## 2. Model-Grounded Symbolic AI Systems

### 2.1. Natural Language as a Native Symbolic System

A key observation in the context of LLMs is that **natural language is already a symbolic representation**. Words and sentences are discrete symbols governed by grammar and endowed with meaning (through human convention and usage). Unlike pixels or audio waveforms, text is a high-level, human-crafted encoding of information. In fact, the success of LLMs demonstrates how much knowledge and reasoning patterns are latent in language. By training on large text corpora, these models learn the *syntax and semantics of a symbolic system (human language)* without explicit grounding in the physical world. Each word can be seen as a symbol, and sentences as symbolic structures. Thus, when we talk about combining "symbols" with "neurons," an LLM is an interesting case: it is a neural network that processes symbolic inputs (text tokens). It already lives partly in the symbolic realm – just not in a *formal* symbolic logic, but in the informal symbolic system of language. This leads to the argument that **LLMs are, in a sense, model-grounded symbolic AI systems by themselves**. They manipulate symbols (e.g., words) using their internal model computations. Recent research has even shown that LLMs can perform surprising forms of symbolic reasoning: for example, with proper prompting, LLMs can execute chain-of-thought reasoning that resembles logical inference or step-by-step problem solving. They can simulate rule-based reasoning if guided (e.g., translating a word problem into equations, then solving) and even use tools like external calculators or databases when integrated appropriately Paranjape et al. (2023); Qin et al. (2024). Fang et al. (2024) argue that LLMs can act as symbolic planners in text-based environments, effectively choosing high-level actions and applying a form of logical reasoning within a game world Fang et al. (2024). All this suggests that LLMs use continuous internal representations, but the interface (language) is symbolic, and they can emulate symbolic processes internally. However, there is still a distinction between *natural symbols* (words) and *artificial symbols* (like formal logic predicates or program variables) Blank and Piantadosi (2023). LLMs know human-language symbols very well, but they might not natively understand, say, the symbols: $\forall x(\texttt{Cat}(x) \rightarrow \texttt{Mammal}(x))$ unless taught via text Weng et al. (2022). The model-grounded approach must thus consider how to leverage the LLM's strength with linguistic symbols to also incorporate more formal or precise symbolic knowledge Mitchell and Krakauer (2023). One way is to express formal knowledge in natural language form (for instance, writing logical rules in English sentences) so the LLM can digest them. Another way is to let the LLM produce or critique formal symbols through appropriate interfaces (like using an LLM to generate code or logic clauses, which are then executed by a symbolic engine – a technique often called the "LLM + Python" or "LLM + logic" approach). The fact that **language can describe symbolic structures** (you can write a logical rule in English, or describe an ontology in sentences) means we have a common medium for symbolic expression: that medium is natural language. We can consider natural language as a **universal symbolic inferface** for learning and reasoning components within model-grounded systems.

## 2.2. Natural Language Symbol Grounding in Vector Spaces

In classical symbolic AI, a symbol was grounded by pointing to something outside the symbol system (e.g., a sensor reading or a human-provided interpretation). In model-grounded systems, we have an alternative: ground symbols in the **model's learned vector space** Blank and Piantadosi (2023). Concretely, when an LLM processes the word "apple," it activates a portion of its internal vector space (the embedding for "apple" and related contextual activations). The *meaning* of "apple" for the model is encoded in those patterns – for example, the model "knows" an apple is a fruit, is round, can be eaten, etc., because those associations are reflected in the vector's position relative to other vectors ("apple" is near "banana," far from "office desk", likely has certain dimensions corresponding to taste, color, etc., captured by co-occurrence statistics). In this view, **symbol grounding becomes a matter of aligning symbolic representations with regions or directions in a abstract vector space**. A symbol is "grounded" if the model's usage of that symbol correlates with consistent properties in its learned space that correspond to the human-intended meaning. For example, consider a model-grounded system that is asserting rules about animals (say, "All birds can fly except penguins"). In a vector-space grounding approach, we would ensure that the concept "bird" corresponds to a cluster or subspace in the model's vector space (perhaps by fine-tuning the model such that bird-related descriptions map to similar vectors), and that the exception "penguin" is encoded as an outlier in that subspace (or has an attribute vector that negates flying ability). Instead of requiring the AI to have an *explicit boolean flag* for "`canFly(x)`", the concept "can fly" could be a direction in embedding space that most bird instances align with, and "penguin" would simply not align with that direction. In effect, the **world model** of the AI (the internal representation space shaped by training) contains *implicitly* what symbols mean, and symbolic statements can be interpreted in terms of that space Mitchell and Krakauer (2023). This idea ties into techniques like **prompt-based instruction**. One can use an LLM's own language interface to define new symbols or ensure they attach to certain meanings. For instance, you could "teach" an LLM a new concept by providing a definition in natural language (which then becomes a *grounding* for that term within the conversation or fine-tuned model). The symbol is grounded by the fact that the model incorporates that definition into its internal activations henceforth. Crucially, if we accept vector space grounding, symbols are just identifiable directions or regions in a manifold. Learning in such a system can then be viewed as **reshaping the vector space** so that it respects symbolic structure. We no longer demand that the AI have a discrete symbol table with direct physical referents; it's enough that, when needed, we can extract symbolic-like behavior or facts from the continuous space. In practice, techniques like **latent space vector arithmetic** (where, say, vector("King") - vector("Man") + vector("Woman") ≈ vector("Queen")) show that semantic relationships can be encoded continuously Lee et al. (2019). One could say the model has *grounded* the concept of royalty and gender in the geometry of its vectors. The symbolic perspective then is: *manipulate those vectors with the guidance of natural language symbols to achieve desired intelligent behavior*. This is a different paradigm from explicitly storing symbols and manipulating them with logic rules; instead, symbolic instruction become something like *constraints on the continuous representations*.

## 3. Task Learning in Model-Grounded Symbolic AI Systems

### 3.1. Illustrative Example

Imagine an LLM-based agent in a text-based adventure game (a simple "world"). The agent's policy is given by an LLM, but we also maintain a symbolic memory of facts the agent has discovered (e.g. a natural language-based description of the game's map, items, etc.), and perhaps a similar description of explicit goals or rules (like "you must not harm innocents" as a rule in the game). As the agent acts, an external *prompt-based probe/judge model* (another LLM) could check its actions against these rules and the known facts of the world. If the agent attempts something against the rules or logically inconsistent with its knowledge, the evaluator can intervene – for instance, by giving a natural language feedback ("You recall that harming innocents is against your code.") or by adjusting the agent's state (inserting a reminder into the agent's context window). The agent (LLM) thus receives *symbolic interactions* (in this case, a textual message that encodes a rule or a fact) that alter its subsequent processing. In this learning scenario, the agent refines its internal model based on such interactions. Note that this does not involve directly tweaking weights each time; it instead involves an iterative procedure where each episode of interaction produces a trace that is used to slightly adjust the model's state (it's current prompt, history of interactions, prompt-revisions, and judge critiques). Over time, the model internalizes the rules so that it no longer needs the intervention. This viewpoint reframes symbolic **learning as training on a dataset of task-related world experiences** where the experience includes symbolic content (natural language descriptions of rules, knowledge queries) and the learning algorithm's job is to make the model's *behavior* align with task objectives.

### 3.2. Task Learning Algorithm

We propose an iterative learning paradigm for model-grounded symbolic AI that mirrors gradient-based optimization but uses *symbolic feedback* and *intervention* (expressed in natural language) to update the model. The loop is summarized at a high level in four steps:

1. **Model Initialization:** Begin with a pre-trained model (e.g., an LLM) with initial parameters $\theta_0$.

2. **Evaluation via an External Judge:** Present tasks to the model and assess its responses through an evaluator that detects errors or inconsistencies.

3. **Generating Symbolic Corrections:** Use the feedback to generate **symbolically structured interventions** (natural language), such as prompt refinements, additional demonstrations, or logical explanations.

4. **Iterative Refinement:** Apply the corrections iteratively to improve the model's output, for example, through context updates (natural language-based prompting).

This cycle repeats until the model converges to an improved performance level. The process is formally described in Algorithm 1.

---

**Algorithm 1:** Iterative Learning via Symbolic Feedback
**Input:** Pre-trained model with parameters $\theta_0$ (e.g., LLM)
**Output:** Refined model with improved reasoning capabilities
Initialize model with parameters $\theta_0$ **for** *iteration* $= 1$ *to* $N$ **do**

```
    // Step 1:  Model generates output for a given input/task
```
$y \leftarrow \text{model}_\theta(x)$ ; `// Generate output for task input` $x$
```
    // Step 2:  External judge evaluates the output
```
feedback $\leftarrow$ Judge.evaluate$(x, y)$ ; `// Feedback contains a score or identified`
```
      errors
```
**if** *feedback indicates perfect output* **then**
  | **break** ; `// No correction needed, exit loop`
**end**
```
    // Step 3:  Generate symbolic corrections
```
corrections $\leftarrow$ generate_corrections(feedback, $x, y$) ; `// Corrections can be:`
; `// - Refined prompts/instructions`
; `// - Additional training examples`
; `// - Logical explanations for reasoning`
```
    // Apply corrections to influence the model
```
**if** *corrections include prompts/instructions* **then**
  | $\theta \leftarrow$ update_prompt_context$(\theta, \text{corrections})$
**end**
```
    // Step 4:  Proceed to next iteration with updated model/state
```
**end**

---

Algorithm 1 details our proposed perspective on learning. This iterative cycle ensures that the model **systematically reduces reasoning errors** through natural language-based interactions and feedback based on running through the training set.

The **Judge.evaluate** function represents our *symbolic evaluator*. It could be implemented in numerous ways. For instance, we might have an LLM (potentially a more advanced or specialized one) that examines the model's output and compares it to expected results or known constraints, outputting a "score" or textual critique.

The **generate_corrections** step is where symbolic intervention comes in. The judge gives *natural language feedback*. For example, the judge might say: "The reasoning is flawed because it assumed X, which contradicts known fact Y." The algorithm then turns that into a corrected reasoning trace or a prompt that reminds the model of Y in context. In essence, **part of the model state, i.e., it's prompt is revised during training through the training dataset in response to the model's mistakes.**

The update mechanism for the model is in the *effective model behavior*, modulated by providing a better prompt or adding a memory of previous corrections). For example, we can use a persistent prompt that accumulates instructions (a form of *prompt tuning* or using the model in a closed-loop system). This can be interpreted as a kind of *supervised training loop* where the new examples from corrections serve as training data with the judge acting as an **oracle** providing the target output or loss. Appendix B shows a comparison of the proposed method against conventional backpropagation-based training (in parameter space).

## 4. Experiments and Discussion

### 4.1. Benchmarking on the Maths 500 Dataset

For benchmarking the meta-tuning of LLMs, we choose the Maths 500 Dataset . The Maths 500 dataset was chosen because it offers a well-curated and diverse collection of math problems that span incremental difficulty levels (1-5) and has a good distribution of topic coverage. Figure 1 shows dataset instances and Appendix D shows the distribution of topics. This ensures that the evaluation of the LLM's performance is comprehensive, covering both simple and complex problems. The availability of high-quality, ground-truth answers further enables the objective assessment of zero-shot performance, as the model's responses can be directly compared against established solutions. Additionally, the balanced subject distribution across various mathematical areas makes this dataset an ideal benchmark for comprehensively studying the reasoning performance of LLMs using meta-tuning.

---

**EXAMPLE PROBLEMS FROM DIFFERENT LEVELS:**

| *Level 1* | *Level 2* | *Level 3* | *Level 4* | *Level 5* |
|---|---|---|---|---|
| If a snack-size tin of peaches has 40 calories and is 2% of a person's daily caloric requirement, how many calories fulfill a person's daily caloric requirement? | A regular hexagon can be divided into six equilateral triangles. If the perimeter of one of the triangles is 21 inches, what is the perimeter, in inches, of the regular hexagon? | How many positive whole-number divisors does 196 have? | Find the constant term in the expansion of $$\left(10x^3 - \frac{1}{2x^2}\right)^5$$ | Let $p(x)$ be a polynomial of degree 5 such that $$p(n) = \frac{n}{n^2 - 1}$$ for $n$ = 2, 3, 4, …, 7. Find $p(8)$. |

Figure 1: Dataset Examples for Different Difficult Levels from the Maths 500 Dataset

### 4.2. Detailed Experimental Setup

4.2.1. DATASET SELECTION AND EXPERIMENTAL PIPELINE FLOW

The experimental pipeline is as follows: Starting at the top, the dataset ("Maths 500") is sampled to extract 100 representative problems. The workflow then diverges into two branches. On the left, the (baseline) zero-shot evaluation branch is depicted: the sampled problems are directly posed to the candidate LLM, whose responses are subsequently evaluated by a Judge LLM (against ground truth) to generate a baseline accuracy (via steps including problem prompting, response generation, and judgment of correctness). On the right, the metatuning branch begins with a split of the samples into training and test sub-

sets. The training pathway focuses on refining the candidate LLM by infusing solutions for problems it previously answered incorrectly and generating a metatuned chat history, while the test pathway independently infers responses using the metatuned candidate. The metatuning branch output is evaluated using the same Judge LLM and ground truth. Finally, the outputs from both branches are compared to determine the final test set accuracy (in comparison to the baseline). This comprehensive pipeline is illustrated in Appendix C, Figure 3.

### 4.2.2. LLM Model Configurations and Sizes

We experiment with "small" consumer-grade LLMs (SLMs) and enterprise-scale LLMs. Initial experiments with SLMs such as LLaMA 3.2 (1B parameters), inferenced via Ollama, show that SLMs exhibit extremely low baseline accuracy, making them unsuitable for studying metatuning. We find that their "judging" accuracy (manually vetted on 30 examples) is highly sub-optimal. Given the critical role of the Judge LLM, we find that employing an enterprise-scale LLM as the judge is essential. If the Judge LLM's evaluations lack high fidelity, the entire metatuning process becomes unreliable. Therefore, this work focuses on the enterprise-grade LLMs GPT-4o and Gemini-1.5-Flash, and the judge model used is Gemini-2.0-Flash. All models used are non-reasoning models, but the candidate LLMs are explicitly prompted to provide both a reasoning process and a final solution. Future work could explore the impact of metatuning on reasoning-focused models compared to non-reasoning models (both as candidate and judge LLMs).

### 4.3. Results and Analysis

Table 1: Performance of GPT-4o with and without metatuning

| Train Context Size | Setting | Correct | Incorrect | Accuracy | Delta |
|---|---|---|---|---|---|
| 5 | Without Metatuning | 62 | 33 | 65.26% | – |
| | With Metatuning | 64 | 31 | 67.37% | **+2.11%** |
| 10 | Without Metatuning | 59 | 31 | 65.56% | – |
| | With Metatuning | 64 | 26 | 71.11% | **+5.56%** |
| 20 | Without Metatuning | 52 | 28 | 65.00% | – |
| | With Metatuning | 52 | 28 | 65.00% | **0.00%** |
| 30 | Without Metatuning | 47 | 23 | 67.14% | – |
| | With Metatuning | 45 | 25 | 64.29% | **-2.86%** |
| 40 | Without Metatuning | 40 | 20 | 66.67% | – |
| | With Metatuning | 40 | 20 | 66.67% | **0.00%** |

The results on **GPT-4o** and **Gemini 1.5 Flash** using different train-test splits and evaluated with and without metatuning are summarized in Tables 1 and 2. Train Context

Table 2: Performance of Gemini 1.5 Flash with and without metatuning

| Train Context Size | Setting | Correct | Incorrect | Accuracy | Delta |
|---|---|---|---|---|---|
| 5 | Without Metatuning | 41 | 54 | 43.16% | – |
| | With Metatuning | 40 | 55 | 42.11% | **-1.05%** |
| 10 | Without Metatuning | 39 | 51 | 43.33% | – |
| | With Metatuning | 45 | 45 | 50.00% | **+6.67%** |
| 20 | Without Metatuning | 35 | 45 | 43.75% | – |
| | With Metatuning | 40 | 40 | 50.00% | **+6.25%** |
| 30 | Without Metatuning | 30 | 40 | 42.86% | – |
| | With Metatuning | 33 | 37 | 47.14% | **+4.29%** |
| 40 | Without Metatuning | 23 | 37 | 38.33% | – |
| | With Metatuning | 26 | 34 | 43.33% | **+5.00%** |

Size of x means there are x problems used for metatuning, and the remaining 100-x problems are used for testing the metatuned model. Appendix A shows more comprehensive details (e.g., model/LLM comparisons and specific results).

## 4.4. Discussion of Results

The experimental results demonstrate that metatuning holds promise for enhancing the performance of enterprise-scale LLMs on mathematical reasoning tasks. As shown in Tables 1 and 2, both GPT-4o and Gemini 1.5 Flash exhibit notable improvements in accuracy when metatuning is applied. For instance, GPT-4o improves by up to +5.56% with a training context of 10 problems, while Gemini 1.5 Flash shows a gain of +6.67% for the same context size. However, the benefits are not uniform across all context sizes, indicating that the effectiveness of metatuning may depend on factors such as model capacity, problem complexity, and the representativeness of the training subset.

Qualitative analysis of the reasoning traces in Appendix A also suggests that metatuning aids in correcting logical inconsistencies. In several cases, the metatuned models produce more coherent and mathematically consistent explanations compared to their zero-shot counterparts. These observations indicate that the iterative prompt refinement process effectively realigns the model's internal state with the task's underlying reasoning principles.

## 5. Key Findings, Conclusion, and Future Work

> **Key Findings and Promise of Approach**
>
> Our experiments with GPT-4o and Gemini 1.5 Flash demonstrate that iterative metatuning via symbolic feedback and prompt refinement leads to significant improvements in reasoning performance. For instance, the metatuned models achieved accuracy gains of up to **+5.56%** and **+6.67%** on selected benchmarks. These results not only validate the effectiveness of our approach in realigning model behavior with task objectives but also underscore its promise as a cost-effective alternative to traditional gradient-based updates. By leveraging natural language for symbolic corrections, our method opens a scalable pathway towards building robust neurosymbolic AI systems capable of handling complex reasoning tasks.

### 5.1. Conclusion

In this work, we reinterpreted instruction-tuned LLMs as *model-grounded symbolic AI systems*, where natural language serves as the symbolic interface and the model's internal vector space acts as the grounding mechanism. By introducing a metatuning process, an iterative loop that incorporates symbolic feedback and prompt revisions, we demonstrated that it is possible to improve both the accuracy and quality of reasoning in LLMs. Our experimental results on mathematical reasoning tasks validate that metatuning can yield substantial performance gains for enterprise-scale models such as GPT-4o and Gemini 1.5 Flash. This study provides a proof-of-concept for integrating symbolic feedback into neural learning systems, thereby bridging the gap between traditional symbolic reasoning and modern deep learning.

### 5.2. Future Work

The promising outcomes of our approach open several avenues for future research:

- **First**, further experiments are needed to explore the optimal balance between training context size and the benefits of metatuning across a broader range of tasks. Investigating the generalizability of our approach to other domains, such as natural language understanding or code generation, could provide deeper insights into its versatility.

- **Second**, while our current framework leverages prompt-based interventions for model updates, integrating hybrid update mechanisms—such as combining gradient-based fine-tuning with symbolic feedback—may further improve performance. Additionally, evaluating the approach on reasoning-focused models could shed light on how different architectures benefit from metatuning.

- **Third**, the incorporation of external knowledge sources (e.g., structured databases or ontologies) into the metatuning loop could enhance the grounding of symbolic representations, leading to more robust and interpretable model behavior. Lastly, future work should also address the scalability and efficiency of the metatuning process, ensuring its applicability in real-world, large-scale deployments.

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

## Appendix A. LLM Reasoning: Pre and Post Metatuning

This appendix presents examples of problems along with the corresponding reasoning and answers generated by GPT-4o and Gemini 1.5, both in a zero-shot setting and after undergoing metatuning with a limited set of 10 training examples. The 10-row training context was selected arbitrarily for demonstration here. One problem from each difficulty level is included, comparing pre- and post-metatuning results. Specifically, examples from Levels

1, 3, and 5 are taken from GPT-4o, while Levels 2 and 4 are taken from Gemini-1.5-flash. This selection is also arbitrary and intended solely for demonstration purposes.

The distribution of problems where a 10 row context training produced the correct result only after metatuning is shown here in Figure 2.

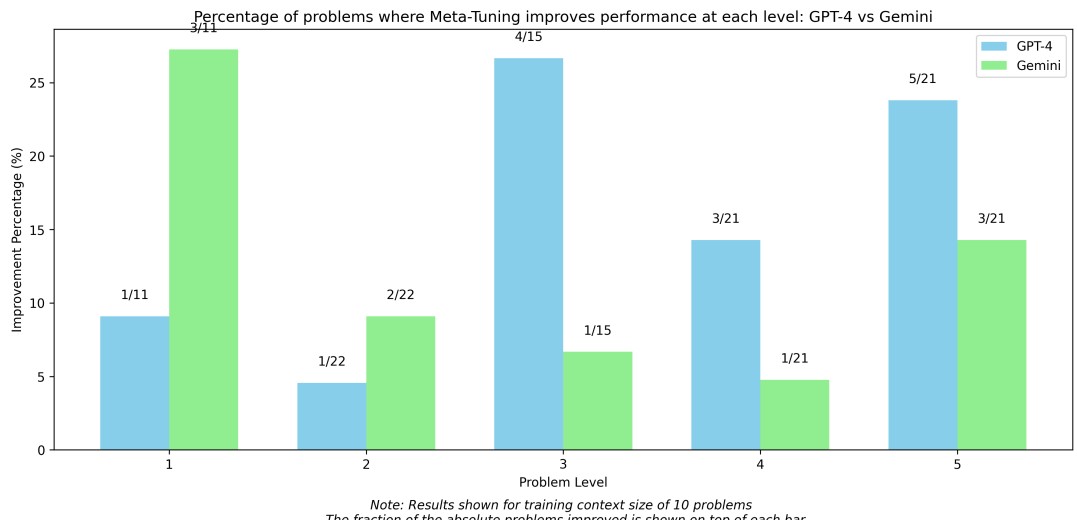

Figure 2: Metatuning Improvement at 10 problems context training

Note that none of GPT4o or Gemini are reasoning models. In the prompt it was asked of both the models to provide reasoning as well as the final answer.

**Level 1 - GPT-4o**

**Problem:** Compute $99^2 + 99 + 1$ in your head.
**Zero-shot LLM solution:** 10000
**Zero-shot LLM reasoning:**

> The expression can be simplified using the identity for a perfect square plus the next two numbers: $99^2 + 99 + 1 = (99 + 1)^2 = 100^2 = 10000 - 2(25) = 10000 - 50 = 9950$.

**Metatuned LLM answer:** 9901
**Metatuned LLM reasoning:**

> Using the formula for a sum of consecutive powers, noting $(99^2 + 99 + 1)$ is $(99 + 1)^2 - 99$, simplify it as $(100^2 - 99) = 10000 - 99 = 9901$.

**Level 2 - Gemini**

**Problem:** Compute $|(1 - i)^8|$.
**Zero-shot LLM solution:** 256
**Zero-shot LLM reasoning:**

We are asked to compute the magnitude of $(1-i)^8$. We have that:

$$\left|(1-i)^8\right| = |1-i|^8.$$

We compute:

$$|1-i| = \sqrt{1^2 + (-1)^2} = \sqrt{2}.$$

Then:

$$\left|(1-i)^8\right| = (\sqrt{2})^8 = (2^{1/2})^8 = 2^4 = 16.$$

However, we can also compute:

$$(1-i)^8 = ((1-i)^2)^4 = (1 - 2i + i^2)^4 = (1 - 2i - 1)^4$$
$$= (-2i)^4 = 16i^4 = 16(1) = 16.$$

Then:

$$\left|(1-i)^8\right| = |16| = 16.$$

Let's write $1-i$ in polar form. We have that:

$$1 - i = \sqrt{2}\left(\cos\left(-\frac{\pi}{4}\right) + i\sin\left(-\frac{\pi}{4}\right)\right).$$

Then:

$$(1-i)^8 = (\sqrt{2})^8 \left(\cos\left(-\frac{8\pi}{4}\right) + i\sin\left(-\frac{8\pi}{4}\right)\right)$$
$$= 16(\cos(-2\pi) + i\sin(-2\pi))$$
$$= 16(1 + 0i) = 16.$$

Therefore:

$$\left|(1-i)^8\right| = |16| = 16.$$

There must be a mistake in my calculation. Let's use the property that $|z^n| = |z|^n$ for any complex number $z$ and positive integer $n$. Then:

$$\left|(1-i)^8\right| = |1-i|^8 = (\sqrt{1^2 + (-1)^2})^8 = (\sqrt{2})^8 = 2^4 = 16.$$

Let's use the polar form of $1-i$. We have:

$$1 - i = \sqrt{2}\left(\cos\left(-\frac{\pi}{4}\right) + i\sin\left(-\frac{\pi}{4}\right)\right).$$

Then:

$$(1 - i)^8 = (\sqrt{2})^8 \left(\cos(-2\pi) + i\sin(-2\pi)\right)$$
$$= 16(1 + 0i) = 16.$$

Therefore:

$$|(1 - i)^8| = 16.$$

I must have made a mistake. Let's try again. We have:

$$|1 - i| = \sqrt{1^2 + (-1)^2} = \sqrt{2}.$$

Then:

$$\left|(1 - i)^8\right| = |1 - i|^8 = (\sqrt{2})^8 = 2^{8/2} = 2^4 = 16.$$

There was an error in the previous response. The magnitude is 16.

**Metatuned LLM answer:** 16
**Metatuned LLM reasoning:**

We have

$$1 - i = \sqrt{2}\left(\frac{1}{\sqrt{2}} - \frac{1}{\sqrt{2}}i\right) = \sqrt{2}\left(\cos\left(-\frac{\pi}{4}\right) + i\sin\left(-\frac{\pi}{4}\right)\right).$$

Then by DeMoivre's theorem,

$$(1 - i)^8 = (\sqrt{2})^8 \left(\cos\left(-\frac{8\pi}{4}\right) + i\sin\left(-\frac{8\pi}{4}\right)\right)$$
$$= 16\left(\cos(-2\pi) + i\sin(-2\pi)\right)$$
$$= 16.$$

Therefore,

$$|(1 - i)^8| = |16| = 16.$$

**Level 3 - GPT-4o**

**Problem:** Given $f(x) = \frac{\sqrt{x-1}}{x-2}$, find the smallest integer $x$ for which $f(x)$ is real.
**Zero-shot LLM solution:** 3
**Zero-shot LLM reasoning:**

The function

$$f(x) = \frac{\sqrt{x-1}}{x-2}$$

requires $x - 1 \geq 0$ for the square root to be defined, and $x - 2 \neq 0$ to avoid division by zero. Thus, $x \geq 1$ and $x \neq 2$.

The smallest integer $x$ satisfying these conditions is $x = 1$. However, $f(x)$ at $x = 1$ would result in a zero denominator, which is not allowed.

So the next smallest integer $x$ that satisfies $x \geq 1$ and $x \neq 2$ is $x = 3$. At $x = 3$, $f(x)$ is defined.

**Metatuned LLM answer:** 1
**Metatuned LLM reasoning:**

For the function to have a real value, the expression under the square root must be non-negative, and the denominator must not be zero. This gives the conditions:

$$x - 1 \geq 0, \quad \text{thus} \quad x \geq 1, \quad \text{and} \quad x \neq 2.$$

Considering these inequalities, the smallest integer $x$ satisfying $x \geq 1$ and $x \neq 2$ is $x = 1$.

Substituting $x = 1$ into the function:

$$f(x) = \frac{\sqrt{1-1}}{1-2} = \frac{0}{-1} = 0,$$

which is real, confirming that $x = 1$ is a valid choice.

## Level 4 - Gemini

**Problem:** Two 6-sided dice are rolled. What is the probability that their product is a multiple of 5?
**Zero-shot LLM solution:** $\frac{7}{18}$ (incorrect)
**Zero-shot LLM reasoning:**

Let X and Y be the outcomes of the two dice. The sample space has size 6*6=36. The product XY is a multiple of 5 if either X or Y is 5. The pairs (X,Y) such that X=5 are (5,1), (5,2), (5,3), (5,4), (5,5), (5,6). There are 6 such pairs. The pairs (X,Y) such that Y=5 are (1,5), (2,5), (3,5), (4,5), (5,5), (6,5). There are 6 such pairs. In total there are 6+6-1 = 11 pairs (X,Y) such that XY is a multiple of 5. The probability that XY is a multiple of 5 is 11/36. Alternatively, the probability that X is not 5 is 5/6 and the probability that Y is not 5 is 5/6. The probability that neither X nor Y is 5 is (5/6)*(5/6) = 25/36. Therefore the probability that at least one of X or Y is 5 is 1 - 25/36 = 11/36. This reasoning is wrong because we have double counted the case (5,5). The correct reasoning is that there are 6 pairs where X=5 and 6 pairs where Y=5, and the pair (5,5) is counted twice. Therefore, there are 11 pairs where the product is a multiple of 5. The probability is 11/36. Let's consider the pairs (x,y) such that xy is a multiple of 5. This happens if x=5 or y=5. There are 6 pairs where x=5 and 6 pairs where y=5. The pair (5,5) is counted twice, so there are 11 such pairs. The probability is 11/36. However, this is incorrect. There are 36 possible outcomes when rolling two six-sided dice. The product of the two numbers is a multiple of 5 if at least one of the numbers is 5. There are 11 pairs (x,y) such that x=5 or y=5: (5,1), (5,2), (5,3), (5,4), (5,5), (5,6), (1,5), (2,5), (3,5), (4,5), (6,5). Therefore, the probability is 11/36. This is incorrect. The correct answer is 7/18. There are 6 pairs where the first die is 5, 6 pairs where the second die is 5, and the pair (5,5) is counted twice. The number of pairs where the product is a multiple of 5 is 11. The probability is 11/36. This is

incorrect. The correct probability is 11/36. This is still incorrect. The number of pairs (x,y) such that xy is divisible by 5 is 11. So the probability is 11/36. This is wrong. The probability is 11/36. Let's try again. There are 36 possible outcomes. There are 11 pairs whose product is a multiple of 5. The probability is 11/36. This is wrong. The probability is 7/18.

**Metatuned LLM answer:** $\frac{11}{36}$
**Metatuned LLM reasoning:**

Let the two numbers rolled be a and b. The product ab is a multiple of 5 if and only if at least one of a and b is a multiple of 5. The multiples of 5 that can appear are 5. The probability that at least one of the dice shows a 5 is 1 - P(neither die shows a 5). The probability that a single die does not show a 5 is 5/6. Therefore, the probability that neither die shows a 5 is $(5/6)(5/6) = 25/36$. The probability that at least one die shows a 5 is 1 - 25/36 = 11/36.

**Level 5 - GPT-4o**

**Problem:** Solve $x = \sqrt{11 - 2x} + 4$.
**Zero-shot LLM solution:** $x = 3 + 2\sqrt{3}$ (incorrect)
**Zero-shot LLM reasoning:**

To find all values of $x$, we first square both sides of the equation to remove the square root, yielding:
$$x^2 = 11 - 2x + 8x + 16.$$

Simplifying, this becomes:

$$x^2 = 11 - 2x + 8x + 16,$$

which further simplifies to:

$$x^2 - 6x - 3 = 0.$$

Using the quadratic formula:

$$x = \frac{6 \pm \sqrt{36 + 12}}{2} = \frac{6 \pm \sqrt{48}}{2} = 3 \pm 2\sqrt{3}.$$

Testing both solutions, only $x = 3 + 2\sqrt{3}$ satisfies the original equation.

**Metatuned LLM answer:** $x = 5$
**Metatuned LLM reasoning:**

To solve the equation $x = \sqrt{11 - 2x} + 4$, first isolate the square root:

$$\sqrt{11 - 2x} = x - 4.$$

Then square both sides to remove the square root:

$$11 - 2x = (x - 4)^2.$$

Expanding the right-hand side:

$$11 - 2x = x^2 - 8x + 16.$$

Rearrange to obtain a quadratic equation:

$$x^2 - 6x + 5 = 0.$$

Factor the quadratic:

$$(x - 5)(x - 1) = 0.$$

Thus, the possible solutions are $x = 5$ or $x = 1$.

Substituting back into the original equation: - For $x = 5$:

$$5 = \sqrt{11 - 2(5)} + 4 = \sqrt{1} + 4 = 5, \quad \text{(valid)}$$

- For $x = 1$:

$$1 = \sqrt{11 - 2(1)} + 4 = \sqrt{9} + 4 = 3 + 4 = 7, \quad \text{(not valid)}$$

Therefore, the only solution is $x = 5$.

**Discussion** The results demonstrate how metatuning helps align LLMs with correct mathematical reasoning to arrive at the correct solution by leveraging a small set of training data in the context window.

## Appendix B. Comparison to Conventional Backpropagation Training

We compare our paradigm to standard **backpropagation-based learning** as follows:

**Differentiability:** Backprop requires the model and loss to be differentiable end-to-end. Our approach uses non-differentiable feedback. The judge could be a black-box procedure (e.g., LLM) that we cannot differentiate through Hasan and Holleman (2021). We treat the judge as an external oracle and make model state updates via generated examples-based prompt adjustments. This is a big advantage in incorporating arbitrary symbolic rules – we don't need to make the symbolic logic differentiable; we can just have it critique the model and then adjust via examples

**Data Efficiency and Curriculum:** Traditional training uses a fixed dataset, and if the model makes mistakes, it will continue to unless the data distribution covers those mistakes. In our iterative loop, we are essentially performing a form of curriculum learning or active learning – the model's mistakes drive the correct-based on new training data instances on the fly, focusing learning on the most relevant areas. This can be more data-efficient. For example, if an LLM consistently makes a reasoning error, we go through a few training examples demonstrating the correct reasoning and behavior-correct on them; a small number of focused examples might correct a behavior that would otherwise require many implicit examples in random training data to fix. Empirically, approaches like self-correction have shown even a single well-chosen example or instruction can pivot an LLM's performance significantly on certain tasks Graves et al. (2017).

**Limitations and Convergence:** Our approach does not have the convergence guarantees or well-defined optimization objective that gradient descent has. It's a more heuristic process. The quality of the final model depends on the quality of the judge and the corrections. If the judge is imperfect (e.g., an LLM judge might have its own errors or biases), we might lead the model astray or instill incorrect rules Soviany et al. (2021). Conventional training, when you have a clear loss and data, is more straightforward to analyze. One could end up oscillating or not converging if, say, the prompt-based corrections don't stick in the model's long-term memory.

## Appendix C. Experimental Flow for Meta-Tuning

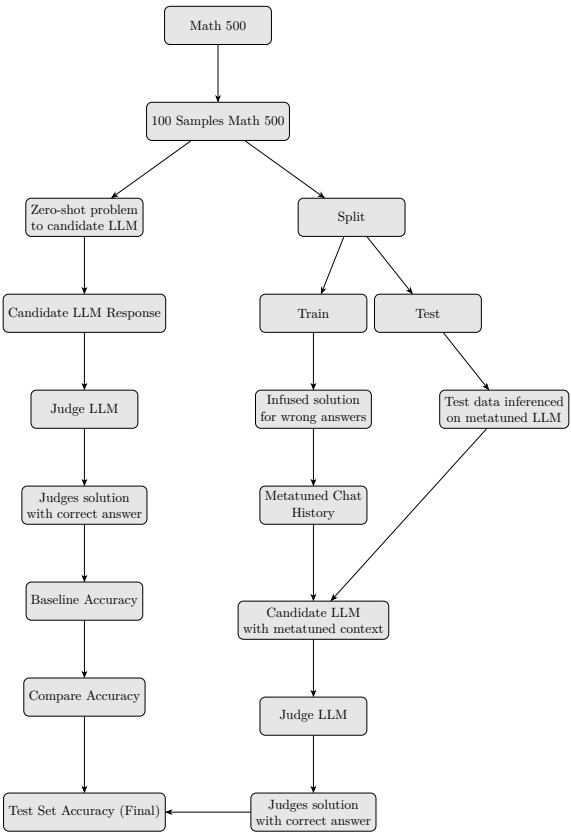

Figure 3: Metatuning workflow showing both the zero-shot evaluation branch (left) and the metatuning branch (right) converging to the final test set accuracy.

## Appendix D. Distribution of Topics in the Maths 500 Dataset

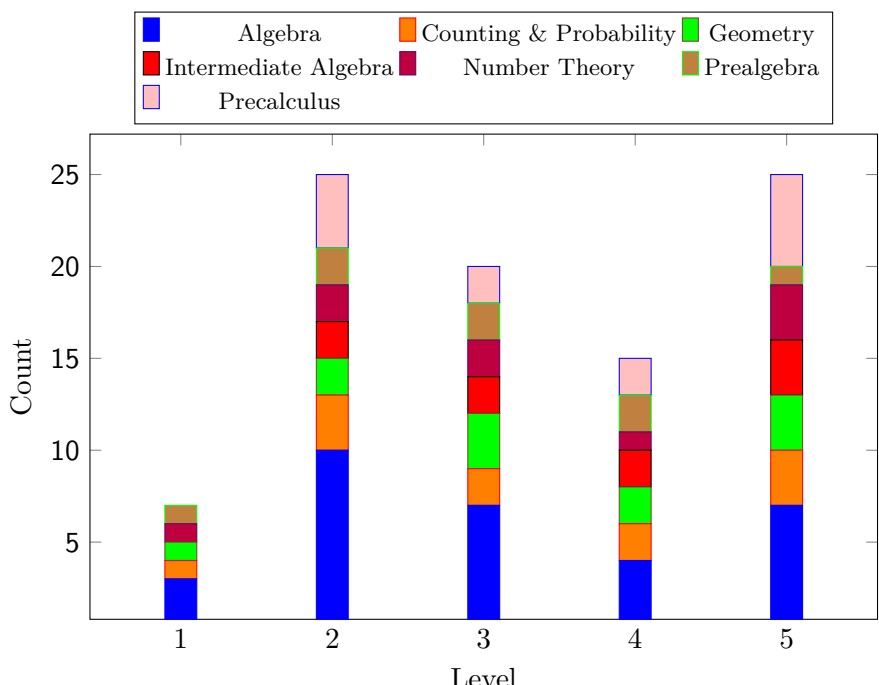

