# OpenReview forum: "Learning and Reasoning with Model-Grounded Symbolic Artificial Intelligence Systems"
_nesyconf.org/NeSy/2025/Conference — NeSy 2025 Poster_

### Official Review · Reviewer_zLiX · 2025-04-01
**In-context prompt tuning reframed as neuro-symbolic learning**

**Rating:** 4
**Confidence:** 4

**Review:**

The paper presents an in-context prompt engineering approach for improving reasoning in large language models (LLMs), reframed as a "neurosymbolic" system, where natural language serves both as the symbolic interface and as a grounding mechanism via "internal vector representations". The idea is that iterative prompt refinement—using symbolic feedback to adjust the model’s "state"—leads toward a more cost-effective alternative to traditional gradient-based updates.

pros:
- the paper is well-written, comprehensible, and very easy to follow
- the point and the experiments are very clear

cons:
- the content reads more like a (rigorous) LLM blogpost rather than a novel research article
- the philosophical and neurosymbolic framings are completely unnecessary
  - i.e., the whole intro (Sec. 1&2) about "the symbolic interface and the model’s internal vector space acts as the grounding mechanism"
is a nice narrative but completely unrelated to what is actually done in the paper
  - i.e., no study of internal representations/grounding is presented, which could be done, e.g., in the sense of mechanistic interpretability [1]

Instead, what is presented is a rather straightforward (in the context of the modern LLM era) iterative (actor-critic) prompt refinement method
- which should be compared as such against the plethora of other in-context (few-shot) prompt tuning techniques [2] or even just base reasoning models.

The paper should then itself be more technically "grounded" in its claims, e.g.
- "This iterative cycle ensures that the model systematically reduces reasoning errors"
  - are there any such (convergence) guarantees?
- "cost-effective alternative to traditional gradient-based updates"
  - that requires actual comparison (it is not necessarily faster)

sentences with wrong/weird wording:
- t, either through context updates (natural language-based prompting).
- part of the model state, i.e., it’s prompt

Overall, I do not unfortunately buy the philosophical/neurosymbolic introductory narrative, and see it more as an obfuscation of a, not-too-original, prompt refinement method rather than a novel neurosymbolic contribution.

[1] Conmy, Arthur, et al. "Towards automated circuit discovery for mechanistic interpretability." Advances in Neural Information Processing Systems 36 (2023): 16318-16352.

[2] Dong, Qingxiu, et al. "A survey on in-context learning." arXiv preprint arXiv:2301.00234 (2022).

**Anonymity:**

Remain anonymous

---

### Official Review · Reviewer_Sbhg · 2025-04-02
**Very interesting idea - preliminary results and algorithm could be expanded**

**Rating:** 7
**Confidence:** 4

**Review:**

This paper reframes instruction-tuned LLMs as model-grounded symbolic systems, as in systems whose symbols are grounded in continuous vector spaces. The authors reach this conclusion by identifying how language is in itself a symbolic language which LLMs are able to process and manipulate through continuous representations to solve tasks. Given this original interpretation of instruction-tuned LLMs, the authors then reinterpret task learning by LLMs as an iterative refinement process, given the interactions between users and the symbolic system via the common medium - language - which is used to steer the model's internal states towards a solution that can solve the task at hand. The authors come up with a simple algorithm for this symbolic refinement of LLMs and apply it to some instruction-tuned LLMs to show improved reasoning capabilities.

The authors support this interpretation by going through a large set of examples in their introduction and background section, making their ideas easily accessible and citing a lot of relevant work. I find their view of grounding the continuous representations of symbolic concepts through symbolic guidance as a refreshing take on the most common views of what neuro-symbolic systems in the context of LLMs are. That said, the algorithm the authors come up with has a big bottleneck that they recognize, which is the LLM judge. I think it is a reasonable first step to try out this approach, but I wonder how feasible it would be to in conjunction teach the judge to improve their instructions, making the learning a two-step mutual improvement (one LLM learns from the other to solve better the tasks while the other one learns better from the first one how to steer it to the right direction). Maybe also learning a judge model in a more classical way would be one of the ways to integrate classic gradient-based learning into the symbolic refinement the authors propose.

The results seem promising albeit I would have been interested in seeing more datasets or maybe different splits of the same dataset to confirm better how good this refinement algorithm really is. I also would have liked to see more systematic studies on the context size, as for example how sensitive the LLM is to the type of problems picked to be "in context". I also wonder if the LLM is more prone to commit the same errors if two similar problems are further away in the chat session, as I think these types of symbolic instructions could be forgotten by the LLM the more the session gets longer and the number of instructions increase.

I was quite confused by the experimental setup, and Appendix C really made it even more confusing for me. I would suggest maybe expanding on it and making this step more explicit, as it is crucial to understanding how the symbolic refinement is integrated.

Finally, I think including some examples of how the corrections provided by the judge make the LLM change their behaviour for a given task would be a great addition to the paper.

In general, I really appreciate the author's idea and I hope to see this preliminary approach and preliminary results expanded in a further submission.

Pros:

- Original idea to reinterpret LLMs and their training in a novel neuro-symbolic way;
- Clearly written and self-contained paper;
- promising preliminary results.

Cons:

- experimental pipeline a bit obscure to me;
- limited experiments.

**Anonymity:**

Disclose identity

---

### Official Review · Reviewer_KXaA · 2025-04-02
**Accept. Small typos to be fixed. Lacks an important comparison.**

**Rating:** 8
**Confidence:** 5

**Review:**

This paper introduces a novel perspective on neurosymbolic AI by reinterpreting instruction-tuned large language models (LLMs) as model-grounded symbolic systems where natural language serves as the symbolic interface. The authors propose "metatuning," an iterative prompt-refinement approach that uses symbolic feedback to enhance LLMs' mathematical reasoning capabilities. Experiments on the Maths 500 dataset demonstrate improvements in reasoning performance across various difficulty levels, with accuracy gains of up to +5.56% for GPT-4o and +6.67% for Gemini 1.5 Flash.

The paper is well-written, and the authors arguments for considering LLMs as model-grounded symbolic systems are backed by related work and sound theory. Section 2 analyses this perspective thoroughly and provides a solid contribution to this dialogue.

The methodology described in this paper is generally sound, with a clear experimental design. The dataset chosen to test the authors' approach is appropriate and explores varying difficulty levels. Additionally, there is proven improvement in specific experimental runs, although no clear pattern of success can be seen (With the gemini model, it is more clear).

A few minor typos and grammatical issues can be easily fixed:
1. In section 3.2, the fourth step "Iterative Refinement" is defined, but the "either through context updates (natural language-based prompting)." is a leading sentence with no "or...."
2. In the first paragraph of section 4.1 the sentence " Figure 1 shows dataset instances and Appendix D) shows the distribution of top-
ics." needs fixing.

A smaller issue is the statement at the beginning of section 4.4 "metatuning can significantly enhance the per-
formance of enterprise-scale LLMs on mathematical reasoning tasks". While the authors have shown empiracally that the Gemini model obtains improved accuracy with meta-tuning (with 10 or more train context size), the GPT-4o model shows too much variance in results to make this statement accurate. Slightly more moderate language is suggested.

The largest issue remains the lack of comparison of the meta-tuning approach , not only with other neurosymbolic approaches, but especially with the Self-Reflection prompting strategy that bears a strong resemblence to the meta-tuning approach. There is a difference with Self-Reflection, in that the proposed approach utilises and external LLM judge with access to ground truths to revise prompts and provide examples. However, with Self-Reflection, the singular LLM critically reviews its own reasoning as its own judge. It would be very useful to compare the two, to show that the use of ground-truth data, and an "oracle" style judge can more aptly address reasoning tasks than Self-Reflection or other prompt engineering strategies (e.g. CoT or ReAct).

I suggest the paper be accepted, as the arguments for considering LLMs as a model-grounded symbolic system is an important contribution to the field of Neurosymbolic AI, and in addition, the methodology of meta-tuning proposed by the authors shows merit with immediate improvements and promise for future applications. The authors should acknowledge the lack of comparison with other prompting strategies, especially Self-Reflection.

**Anonymity:**

Disclose identity